# Negative Regulation of Serine Threonine Kinase 11 (STK11) through miR-100 in Head and Neck Cancer

**DOI:** 10.3390/genes11091058

**Published:** 2020-09-08

**Authors:** Gabriela Figueroa-González, José F. Carrillo-Hernández, Itzel Perez-Rodriguez, David Cantú de León, Alma D. Campos-Parra, Antonio D. Martínez-Gutiérrez, Jossimar Coronel-Hernández, Verónica García-Castillo, César López-Camarillo, Oscar Peralta-Zaragoza, Nadia J. Jacobo-Herrera, Mariano Guardado-Estrada, Carlos Pérez-Plasencia

**Affiliations:** 1Unidad Multidisciplinaria de Investigación Experimental Zaragoza (UMIEZ), Facultad de Estudios Superiores Zaragoza, Universidad Nacional Autónoma de México, Mexico City 09230, Mexico; gabufg@gmail.com; 2Unidad de Investigación Biomédica en Cáncer, Laboratorio de Genómica, Instituto Nacional de Cancerología, Mexico City 14080, Mexico; josejosecarr@gmail.com (J.F.C.-H.); perez.itzel3e@gmail.com (I.P.-R.); dfcantu@gmail.com (D.C.d.L.); adcamposparra@gmail.com (A.D.C.-P.); maga94@comunidad.unam.mx (A.D.M.-G.); jossithunders@gmail.com (J.C.-H.); 3Unidad de Investigación Biomédica en Cáncer, Laboratorio de Genómica del Cáncer, Facultad de Estudios Superiores Iztacala, Universidad Nacional Autónoma de México, Tlalnepantla 54090, Edo.Mex, Mexico; veronica_garcia_367@hotmail.com; 4Posgrado en Ciencias Genómicas, Universidad Autónoma de la Ciudad de México, Mexico City 09790, Mexico; genomicas@yahoo.com.mx; 5Dirección de Infecciones Crónicas y Cáncer, Centro de Investigación Sobre Enfermedades Infecciosas, Instituto Nacional de Salud Pública, Cuernavaca 62100, Morelos, Mexico; operalta@insp.mx; 6Unidad de Bioquímica, Instituto Nacional de Nutrición y Ciencias Médicas, Salvador Zubirán, Mexico City 14000, Mexico; nadia.jacobo@gmail.com; 7Laboratorio de Genética, Licenciatura en Ciencia Forense, Facultad de Medicina, Universidad Nacional Autónoma de México, Mexico City 04360, Mexico; mguardado@cienciaforense.facmed.unam.mx

**Keywords:** microRNA, mir-100, *STK11*, promoter methylation, head and neck cancer

## Abstract

Background: Serine Threonine Kinase 11 (STK11), also known as LKB1, is a tumor suppressor gene that regulates several biological processes such as apoptosis, energetic metabolism, proliferation, invasion, and migration. During malignant progression, different types of cancer inhibit STK11 function by mutation or epigenetic inactivation. In Head and Neck Cancer, it is unclear what mechanism is involved in decreasing STK11 levels. Thus, the present work aims to determine whether STK11 expression might be regulated through epigenetic or post-translational mechanisms. Methods: Expression levels and methylation status for STK11 were analyzed in 59 cases of head and neck cancer and 10 healthy tissue counterparts. Afterward, we sought to identify candidate miRNAs exerting post-transcriptional regulation of STK11. Then, we assessed a luciferase gene reporter assay to know if miRNAs directly target STK11 mRNA. The expression levels of the clinical significance of mir-100-3p, -5p, and STK11 in 495 HNC specimens obtained from the TCGA database were further analyzed. Finally, the Kaplan–Meier method was used to estimate the prognostic significance of the miRNAs for Overall Survival, and survival curves were compared through the log-rank test. Results: STK11 was under-expressed, and its promoter region was demethylated or partially methylated. miR-17-5p, miR-106a-5p, miR-100-3p, and miR-100-5p could be negative regulators of STK11. Our experimental data suggested evidence that miR-100-3p and -5p were over-expressed in analyzed tumor patient samples. Luciferase gene reporter assay experiments showed that miR-100-3p targets and down-regulates STK11 mRNA directly. With respect to overall survival, STK11 expression level was significant for predicting clinical outcomes. Conclusion: This is, to our knowledge, the first report of miR-100-3p targeting STK11 in HNC. Together, these findings may support the importance of regulation of STK11 through post-transcriptional regulation in HNC and the possible contribution to the carcinogenesis process in this neoplasia.

## 1. Introduction

Head and Neck cancer (HNC) is the sixth most common cancer worldwide, and approximately 200,000 annual deaths are attributed to this type of cancer [1]. Head and neck squamous cell carcinoma comprises more than 95% of all HNC [2]. In addition, HNC is a heterogeneous disease affecting different anatomical areas such as oral cavity, oropharynx, hypopharynx and larynx [3]. It has been established that tobacco, alcohol, and their synergistic effect are strong risk factors for HNC development [4,5]; nevertheless, the presence of high-risk human papillomaviruses (HPVs) has also been reported, and they are considered to be etiological agents for some of these cancers [6].

As previously mentioned, this malignancy is determined by several alterations and risk factors, which in combination lead to greater tumor heterogeneity and thus pose a complex challenge for treatment [7]. There are molecular markers, such as HPV-positivity, that could indicate increased sensitivity to some chemotherapeutics such as roscovitine [8]. Despite this, it is of great interest in the research and clinical fields to fully understand the molecular events that generate this carcinogenic process and to clearly identify the key genes at the center of its regulation; this knowledge is imperative in the quest for novel therapeutic approaches and therapies.

Liver Kinase B1, LKB1 (coded by STK11 gene), is a tumor suppressor gene that regulates several biological pathways such as p53-dependent apoptosis, energetic metabolism, fatty acid biosynthesis, proliferation, and cell cycle polarity [9,10,11,12]. Moreover, LKB1 phosphorylates and activates AMP-activated protein kinase (AMPK). The LKB1/AMPK pathway negatively regulates cancer cell proliferation and metabolism, and is also involved in tumor invasion and migration, as an important carcinoma progression hallmark [13]. Human STK11 gene is located on chromosome 19p13.3 [14]; loss-of-function mutations in its coding sequence have been found in lung [15,16], breast [17,18], cervical [15], pancreatic and biliary carcinomas [15,19,20], and also in testicular cancer, malignant melanoma [21,22], hepatocellular carcinoma [23], and HNC [24]. Recent reports suggest that STK11 might play an important role in tumor cell proliferation and invasion capacity through regulation of p53 and p21/WAF1 expression [25]. Hence, STK11 could control the regulation of important hallmarks of cancer such as cellular energetics and cell proliferation, among others.

The function of LKB1 is regulated by diverse mechanisms in different types of tumors, with loss of heterozygosity LOH [26], somatic mutations that inactivate the function of the protein (for an extensive review [27]), and hyper-methylation in the promoter region of the gene [28,29] having been observed. Regarding the negative regulation exerted by microRNAs on STK11 expression in human cancers, it has been reported that miR-199a, miR-17 and miR-155 might regulate STK11 expression in cervical cancer [30,31,32]; in pancreatic cancer mir-7 represses autophagy via directly targeting LKB1 [33]. Remarkably, despite the role that STK11 plays in regulating various hallmarks of cancer there is little information on the inhibition afforded by microRNAs. In the present study, we investigate the epigenetic and post-transcriptional mechanisms involved in controlling the expression of STK11 in HNC. To our knowledge, this is the first report suggesting evidence that downregulation of STK11 in HNC is not attributable to gene promoter methylation, but to post-transcriptional regulation in which miR-100-3p had a negative regulation on LKB1 mRNA level through the union to its 3′ UTR region. Finally, data obtained from the TCGA expression database showed a significant correlation between STK11 expression and overall survival in HNC.

## 2. Materials and Methods

### 2.1. Clinical Samples

Our study included 59 patients prospectively enrolled in the National Cancer Institute of Mexico (Instituto Nacional de Cancerología, INCan) tumor bank protocol. All patients signed informed consent; the protocol No. 014/003/CCI was approved by the Institutional Ethics Committee (CEI/892) following the Declaration of Helsinki. Immediately after surgical excision, tumor biopsies were split into two pieces, one for pathological confirmation of at least 80% of tumor cells and the other for nucleic acid isolation (RNA and DNA). Biopsies for nucleic acid isolation were frozen in liquid nitrogen until DNA and RNA extraction. Eligibility criteria were (1) patients with a confirmed pathological diagnosis of head and neck cancer stages II to IV; (2) histology of squamous cell carcinoma; (3) no previous oncological treatment. Normal samples were obtained from non-pathological adjacent tissue.

### 2.2. Nucleic Acid Isolation and Assessment

Tissues were homogenized using GREEN Bead Lysis (Bullet Blender-Next Advance, Troy, NY, USA) in the MagNA Lyser Instrument (Roche Diagnostics, Basilea, Suiza). DNA isolation was done with QiAmp DNA Blood MiniKit (Qiagen, Hilden, Germany). RNA was isolated using TRizol Reagent (Ambion, Austin, TX, USA) according to the manufacturer’s instructions. Samples were stored at −80 °C until used. To determine specimen adequacy for PCR amplification, human ß-globin was employed as an internal control.

### 2.3. HPV Detection and Genotyping

HPV detection and genotyping were performed as previously described by Sotlar et al. [34] with little modification. First PCR was performed with 100 ng of DNA as a template and using the PCR Master Mix (Thermo Scientific, Waltham, MA, USA) in a total volume of 20 µL. Reactions (GP-E6/E7) were subjected to one 95 °C for 3 min, then 40 cycles of 95 °C for 30 s, 38 °C for 1 min and 72 °C for 30 s, followed by 72 °C for 10 min in an Arktik PCR machine (ThermoFisher, Waltham, MA, USA). Nested PCR containing 1.5 µL of first PCR product reaction was performed with the same PCR Master Mix. Two primer cocktails were used for genotyping: cocktail 1 for HPV-16, -18, -31, -59, and -45; and cocktail 2 for HPV-33, -6/11, -58, -52, and -56. Reactions were subjected to the following amplification conditions: 95 °C for 3 min, then 38 cycles of 95 °C for 30 s, 56 °C for 30 s, 72 °C for 30 s, followed by 72 °C for 4 min. DNA isolated from SiHa cell line was used as an internal control for primer cocktail 1, as it contained integrated human papillomavirus type 16 genome; and 5 pg of HPV-33 plasmid was used as a control for primer cocktail 2. ß-globin was used as a housekeeping gene: (forward 5′- GGTTGGCCAATCTACTCCCAGG-3′ reverse 5′- TGGTCTCCTTAAACCTGTCTTG-3′) with the following conditions: 95 °C for 3 min, then 35 cycles of 95 °C for 30 s, 59 °C for 30 s, 72 °C for 30 s, followed by 72 °C for 10 min. Amplicons obtained from all PCRs protocols were subjected to 2% agarose gel electrophoresis along with positive and negative controls and 100 bp DNA ladder, stained with ethidium bromide. Images were processed using Gel Doc RZ Imager (BioRad, Hercules, CA, USA). Sequences of all the type-specific nested PCR primers are shown in Table 1.

### 2.4. Bisulfite Conversion and STK11 Gene Methylation

Primer sequences for methylation-specific PCR (MSP) of STK11 were designed using the MethPrimer Software 2.0 (http://www.urogene.org/cgi-bin/methprimer/methprimer.cgi) [35]. The CpG island enriched DNA sequence near the STK11 promoter was obtained using the GenomeBrowser island prediction function (chr19:1205041-1207555); the DNA sequence was uploaded to the MethPrimer program in order to obtain MSP primers. Finally, we chose a primer set that covered a 180 pb region near to the STK11 promoter. For bisulfite conversion, 500 ng of DNA were treated with EZ DNA Methylation-Direct Kit (Zymo Research, Irvine, CA, USA). Then, 20 ng of the converted DNA were used for methylation-specific PCR and performed by using the STK11 methylated (forward 5′-CGA GTT TTA TCG AGG TTA TAG TCG T-3′, reverse 5′-GTT AAT TAA ACC TAC CAT CCC CG-3′) and unmethylated (forward 5′-TGA GTT TTA TTG AGG TTA TAG TTG T-3′, reverse 5′-ATT AAT TAA ACC TAC CAT CCC CAA C-3′) primer set. Human methylated and non-methylated DNA sets were used as positive and negative control respectively (Zymo Research, Irvine, CA, USA). The MSP reactions were performed using the Zymo Taq DNA Polymerase Kit (Zymo Research, Irvine, CA, USA) with the following amplification conditions: 95 °C for 10 min, then 40 cycles of 95 °C for 30 s, 60 °C for 35 s, 72 °C for 35 s, followed by 72 °C for 7 min in an Arktik PCR machine (ThermoFisher, Waltham, MA USA). PCR products were analyzed by 2% agarose gel electrophoresis, stained with ethidium bromide, and documented using Gel Doc RZ Imager (BioRad, Hercules, CA, USA).

### 2.5. Real-Time PCR Analysis

STK11 mRNA expression complementary DNA (cDNA) was obtained using High-Capacity cDNA Reverse Transcription kit (Applied Biosystems, Foster City, CA, USA) and assessed by real-time PCR with a qPCR Master Mix (Thermo Fisher, CA, USA) following the manufacturer’s instructions. The sequence of primers was as follows: for STK11, Forward 5′-ACG GTG CCC GGA CAG G-3′, reverse 5′-CTG TGC CGT TCA TAC ACA CG-3′; and for ß-actin, Forward 5′-ATG ACT TAG TTG CGT TAC ACC CT-3′, Reverse 5′-TGC TCG CTC CAA CCG ACT G-3′. Triplicate RT was performed for each assay; data for mRNA expression were normalized with a ß-actin housekeeping gene. The comparative ΔΔCt method was used to quantify gene expression, and the relative quantification was calculated as 2ˆ−(ΔΔCt). To measure expression levels of STK11, 36 tumor and 10 normal adjacent samples were used.

MiRNA target prediction was assessed using miRWalk 2.0 (http://zmf.umm.uni-heidelberg.de/mirwalk2 [36], and RNAhybrid 2.2 (http://bibiserv.techfak.uni-bielefeld.de/rnahybrid [37], where we found that miR-17-5p, miR-100-3p, miR-100-5p, and miR-106a-5p had putative target regions in the STK11 3′UTR. Relative expressions of mature sequences of miR-17-5p, miR-100-3p, miR-100-5p, and miR-106a-5p were quantified by using the TaqMan Universal Master Mix II kit and the appropriate miRNA TaqMan probes (Applied Biosystems, Foster City, CA, USA). Briefly, cDNA was generated from 100 ng total RNA with the TaqMan Micro-RNA Reverse Transcription Kit (Applied Biosystems) in a 15 µL volume; qPCR was performed using 1 µL cDNA and the miRNAs taqman probes (Applied Biosystems). Amplification conditions were 10 min at 95 °C, followed by 40 cycles of 95 °C for 15 s, 68 °C for 60 s. Triplicate RT was performed for each assay; relative expression data was calculated through the 2ˆ−(ΔΔCt) method (Applied Biosystems) and normalized relative to U6 snRNA. To measure expression levels of the above-mentioned miRNAs, 20 tumor samples and 10 healthy controls were used.

### 2.6. Cell Culture and Transfection

Hela cell line (ATCC, CCL-2) was used as a model of interaction between STK11 mRNA 3′ UTR and putative microRNAs and was maintained following ATCC recommendations.

All plasmids and microRNA mimics used for this study were transfected using Plus Reagent-supplemented Lipofectamine 2000 transfection reagent (Invitrogen, Carlsbad, CA, USA), following the manufacturer’s protocol.

### 2.7. Luciferase Reporter Assay

Reporter plasmids were constructed by ligation of synthetic oligonucleotide duplexes (IDT, San Jose, CA, USA) containing putative miR-100-5p and miR-100-3p target regions in the STK11 3′UTR, obtained from RNAhybrid 2.2, into the pMIR-REPORT System (Ambion, Austin, CA, USA) to form a DNA duplex with overhanging SpeI and HindIII half sites in the 5′ and 3′ ends, respectively, which was cloned into the appropriately digested pMIR-REPORT plasmid. This construct was co-transfected with miR-100-5p and miR-100-3p mirVana miRNA mimic (Applied Biosystems, Foster City, CA, USA) and the pMIR-REPORT and renilla Control Plasmid (Ambion, Austin, TX, USA) into Hela cells. Luciferase activity was analyzed using the Dual-Luciferase Reporter Assay System (Promega, Madison, WI, USA) 48 h after transfection, in a GloMax 96 Microplate Luminometer (Promega). Luciferase activity was normalized to β-gal activity for each transfected well; each experiment was performed in triplicate. As negative controls we employed a miRNA random sequence (scrambled) and mutated sequences of STK11 (three nucleotides changed). Three independent experiments were performed, and the data are presented as the mean +/− SD.

### 2.8. Patient Survival Analysis of STK11, miR-100-3p, and miR-100-5p

To know the clinical significance of STK11, miR-100-3p, and -5p expression levels, the TCGA database was analyzed. The expression data of 495 HNSC specimens were obtained, normalized, and log2 transformed. Finally, Cox regression analysis and the Kaplan–Meier method were used to estimate the prognostic significance of the miRNAs for overall survival, and survival curves were compared through the log-rank test. The median value of each RNA expression was defined as a cut-off value between high expression and low expression. Statistical significance was defined as *p* < 0.05.

### 2.9. Statistical Analysis

All values are expressed as the mean ± SD. Statistical analysis between normal and tumor samples were performed with a two tailed Student’s-*t* test. Regarding clinical samples, a two-way ANOVA of the clinicopathologic data was performed using SPSS Statistics v. 22.0 followed by Tukey’s multiple comparison test. The clinical characteristics used in the ANOVA were MSP status, anatomic region, and clinical stage. Correlation analyses were performed using the spearman correlation. *p*-Value < 0.05 was considered as significant.

## 3. Results

### 3.1. STK11 mRNA is Down-Regulated in HNC

The studied population consisted of HNC tumors from 59 patients (Table 2 summarizes clinicopathologic and data from patients) of INCan, Mexico. Patient ages ranged from 28 to 88 years (median 65 years). According to this study, 58% of patients (*n* = 35) have a history of alcoholic habit and cigarette smoking; smokers were defined by those subjects who consumed more than 15 cigarettes a day for a period of time ≥15 years; while alcoholism was defined as consumers of ≥4 whisky equivalents daily (30 cc for each equivalent) for ≥15 years. Primary tumor was located in the lip and oral cavity (*n* = 50, 84.7%), pharynx (*n* = 7, 11.9%), larynx (*n* = 1, 1.7%) and nasal cavity (*n* = 1, 1.7%). Almost ninety percent of patients (54/59) presented advanced stages III or IV of HNC, whereas 5/59 patients (~9%) presented early stages (II). Patient gender distribution was male (41/59), and 32% were female (19/59). The histology of all samples was squamous cell carcinoma. The presence of high-risk HPV-16 was detected in 4/59 (6.8%) samples by multiplex-PCR; these four samples (2-IVA and 2-IVC) were used in all experiments. Interestingly, other HPV genotypes analyzed were negative in all samples (HPV-16, -18, -31, -59, -45, 33, -11, -58, -52, and -56).

To determine the expression level of STK11 in samples, a qRT-PCR analysis was performed using 36 HNC tumors and 10 non-pathological controls (adjacent healthy tissue). Interestingly, the STK11 expression level diminished ∼95% in HNC tumors samples analyzed compared to samples obtained from adjacent healthy tissue (Figure 1). Four tumor samples from stage II (early stage), and six and twenty-six HNC tumors classified as III and IV stages (advanced stages), respectively, were analyzed to quantify STK11 expression level (Figure 1a). Accordingly, the expression level was significantly reduced in all analyzed stages compared to normal tissue. While analyzing matched samples of adjacent healthy tissue and tumor counterparts, the same pattern of subexpression of STK11 is observed (Figure 1b).

### 3.2. STK11 Promoter Methylation Status

As inactivation of tumor suppressor genes can occur via promoter methylation, we assessed the status of the STK11 promoter methylation in HNC samples through MSP. Accordingly, the STK11 promoter was found to be demethylated or partially methylated in the majority of tumor samples (Figure 2); only two tumor tissues were found to be methylated. These results suggested that a post-transcriptional mechanism could negatively regulate STK11 expression. Next, we performed an ANOVA between the STK11 expression and the MSP status, clinical stages and histological grade. As expected, we observed significant differences between the STK11 promoter methylation status and the expression of STK11 (Figure 2c), although no significant difference was found in the STK11 expression level in early or advanced stages of HNC tumors, their histological grade nor between the anatomic region (Appendix A), suggesting that the down-regulation of STK11 occurs since early stages and is maintained during the progression of the disease.

### 3.3. miR-100-3p Interacts with STK11 mRNA

It has been established that there are miRNAs that down-regulate virtually every expressed gene by means of post-transcriptional regulation. We searched public databases for microRNAs that could interact with the 3′-UTR of STK11; thereby, we found that miR-17-5p, miR-106a-5p and miR-100-3p and -5p might be implicated in STK11 down-regulation in HNC, since STK11 could be a putative target. Then, we assessed expression levels of those miRNAs in tissue samples from HNC patients and found no variation in miR-17-5p and miR-106a-5p (Figure 3a,b) between tumor samples and their healthy counterparts. However, the overexpression of miR-100-5p and miR-100-3p was observed in the same samples (Figure 3c,d). The same was observed when the expression of each miRNA was analyzed among adjacent normal tissues matched with tumor tissues (Figure 4). Similarly to STK11, we observed no difference in the expression of the miRNAs when comparing the clinical stage (early vs. advanced) and their histological grade (0–4); this suggests that their up-regulation is maintained during the progression of the disease (Appendix A).

Therefore, to confirm whether miR-100-5p and miR-100-3p exerted a direct regulation on mRNA of STK11, we performed a Luciferase reporter assay in Hela cells transfected with the miR-100-5p and miR-100-3p mimics. As shown in Figure 5, luciferase activity decreased significantly after the transfection of the mimic in just one of the interacting regions (miR-100-3p), with it being the most efficient interaction site at decreasing luciferase levels. These experiments confirm the predicted binding of miR-100-3p with the 3′-UTR of STK11. Additionally, the experiment was performed using a mutant interaction site with three changes in the seed sequence (Figure 5). The observed rescue of the luciferase emission confirmed the specificity of the miR-100-3p and STK11 3′-UTR interaction. Moreover, we correlated the expression of STK11 and the two strands of miR-100 (Figure 5b,c). In this analysis, we observed a significant negative correlation between STK11 and miR-100-5p, which is in agreement with our hypothesis. Then, we assessed the clinical relevance of these molecules using data from the TCGA Head and Neck project [38,39]. The Kaplan–Meier analysis showed that only STK11 expression levels were significantly associated with the clinical outcome (Figure 6) and that its expression was highly heterogeneous in normal vs. tumor samples, although this differences were not significant while miR-100-3p and -5p showed no association with the clinical outcome.

## 4. Discussion

While previous epidemiological studies have reported that HPV is found in HNC tumors [6,40,41,42,43,44,45], our data showed little evidence that HPV is etiologically linked to this kind of carcinoma, probably because most of our cases correspond to oral cavity carcinoma. We found HPV in 6.8% of the tumor samples, and all of them corresponded to the IV-A to IV-C clinical stage. These findings are statistically consistent with those reported by Gillison and colleagues [46] in HPV-infected HNC patients.

Worldwide, common high-risk HPVs are HPV-16, -18, -66, and -51, among others [46], and according to our data, HPV-16 was found in four HNC samples. Even though HPV-18 is also very common in cervical cancer, our HNC results correlate with a previous systematic review where authors explain the rare presence of HPV-18, aside from other oncogenic HPVs [47]. In the present study, by means of two primer cocktails, different HPV genotypes were analyzed (cocktail 1 for HPV-16, -18, -31, -59, and -45, and cocktail 2 for HPV-33, -6/11, -58, -52, and -56). Although the number of cases included in the present study was not sufficient to make epidemiological conjectures about the presence of HPV in oral cavity cancer in the Mexican population, the low frequency of viral infection could shed some light on the involvement of additional etiological factors in the development of this type of tumor, aside from the fact that the most frequent primary origin in our series was the oral cavity. More studies involving a broader patient cohort will be needed in order to determine the HPV distribution in HNC tumors in Mexico.

It has been recently shown that STK11 is a tumor suppressor gene associated with Peutz-Jeghers syndrome, which is a multiple-cancer susceptible disease characterized by inactivation of LKB1 [48]. Somatic loss of function STK11 mutations has been found in several cancer types. Additionally, STK11 is downregulated in tumor cells, and this behavior is thought to be part of the tumorigenesis processes and the higher rate of cell proliferation [49,50]. STK11 exerts its functions by phosphorylating and activating AMPK; then, LKB1/AMPK controls and restrains cell metabolism and proliferation through the inhibition of biosynthetic pathways such as lipid, glycogen, and rRNA biosynthesis. LKB1/AMPK pathway can also inhibit cell proliferation through the inactivation of the mTOR pathway; hence, activation of AMPK by LKB1 suppresses cell growth and proliferation when energy and nutrient levels are limited [51].

In the present study, we showed a down-regulation of STK11 expression in HNC tumors, compared to their normal counterparts. Then, we searched for a mechanism that could explain the decrease in the levels of STK11 expression in tumor biopsies. The methylation of CpG islands in promoter regions drives the inactivation of gene expression in different tumor suppressor genes. We were not able to find a correlation between methylation and expression levels of STK11 in tumor samples through our MSP methylation status analysis. Since no significant statistical difference or relation between STK11 expression, methylation status, and clinicopathological features was found in our study, it can be assumed that STK11 downregulation is not the result of promoter methylation. Nonetheless, the methylation status of the STK11 promoter region may be associated with a decrease in expression levels observed in tumor samples; it would be necessary to analyze the entire region that covers 2500 pb. Moreover, although MSP is the most common technique used for DNA methylation assays, it should take into consideration for further experiments the use of a quantitative approach [52].

It has been shown that in a broad range of tumor types, methylation status might change compared to non-cancerous tissues [51,53], and that it can be used as a potential diagnostic tool. Some evidence notes the difference between methylation status in cancer vs. healthy tissues, such as the case of hepatocellular carcinoma (HCC) and the frequently methylated 3OST2 gene. Interestingly, the authors also reported that STK11 was rarely methylated in HCC samples [54] and, according to our results, the tendency of STK11 to be unmethylated or partially methylated was retained in HNC tumors. Concordantly, Ekizoglu and colleagues demonstrated that STK11 promoter is partially methylated in HNC; their study showed no significant difference between normal and tumor tissues [50]. In this context, our results also showed that the methylation status was independent of HPV presence. Although previous reports have assessed genome-wide methylation and expression differences in HPV (+) or HPV (−) HNC, finding that promoter hypermethylation is widely recognized as an HNC progression mechanism [50], we report here that the methylation status for the promoter of STK11 gene is not related to the HPV presence or clinical stage. Further studies are necessary to determine the involved mechanism in STK11 demethylation in HNC, as well as the point at which this mechanism is consistent between HPV (+) and HPV (−) tumors. Interestingly, HPV-positive HNC had a different miRNA profile in comparison to HPV-negative HNC [55].

As these results suggested that a post-transcriptional mechanism instead of methylation status negatively regulated expression of STK11, we searched for putative miRNAs regulating 3′ UTR of STK11. We found that miR-100-3p was able to regulate STK11 and that it was overexpressed in tumor samples, this means that the more miR-100-3p is present, the lower expression of STK11 will be, as can be appreciated in Figure 5.

miR-100 is deregulated in several cancer types and acts as a tumor suppressor, as is the case of oral cancer [56], epithelial ovarian cancer [57], bladder cancer [58], and hepatocellular carcinoma [59], among others, but to our knowledge, this is the first report showing STK11 is negatively regulated by miR-100-3p in human cancer. Interestingly, miR-100 was found to decrease its expression in patient biopsies obtained from oral cavity tumors (OSCC) and cell lines derived from these cancers. It is possible that the experimental design could induce the observed differences in miR-100 expression levels. In our study, we compared tumor tissue versus adjacent normal tissue, while the study by Henson et al. made use of primary cultures of keratinocytes obtained from the oral cavity of healthy subjects as controls [56]. The experimental variability induced by culture conditions should be analyzed in more detail. Specifically, we found that STK11 under-expression in HNC could be caused by direct interaction with miR-100-3p but there should be other mechanisms including different microRNAs explaining reduction of mRNA levels in HNC.

## 5. Conclusions

STK11 is a multifunctional gene associated with several types of cancer, including HNC. We report that in HNC, STK11 is downregulated regardless of tumor stage or HPV presence (6.8% of total tumors), although mir-100-3p shows marginal expression in the TCGA database; we found that the reduction in expression levels of STK11 could be explained in part by the negative regulation exerted by miR-100-3p. Finally, our data established miR-100-3p to be a bona fide negative regulator of STK11, and a potential role for miR-100-3p as an oncomiR in this type of cancer via the inactivation of tumor suppressor STK11 can be proposed, although further assays may be necessary. To our knowledge, this is the first report of an oncogenic role of miR-100-3p.

## Figures and Tables

**Figure 1 genes-11-01058-f001:**
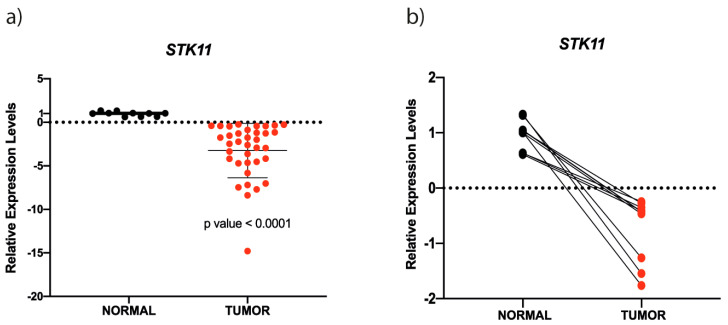
*STK11* is down-expressed in HNC samples. (**a**) Expression levels of *STK11* determined by qRT-PCR in HNC samples (red circles) and healthy tissue samples (black circles). *STK11* was 5-fold down-expressed in HNC tumors compared to control tissue. The statistically significant difference is indicated by *p*-values (*p*-value < 0.0001). (**b**) Matched expression of healthy adjacent tissue and its tumor counterpart.

**Figure 2 genes-11-01058-f002:**
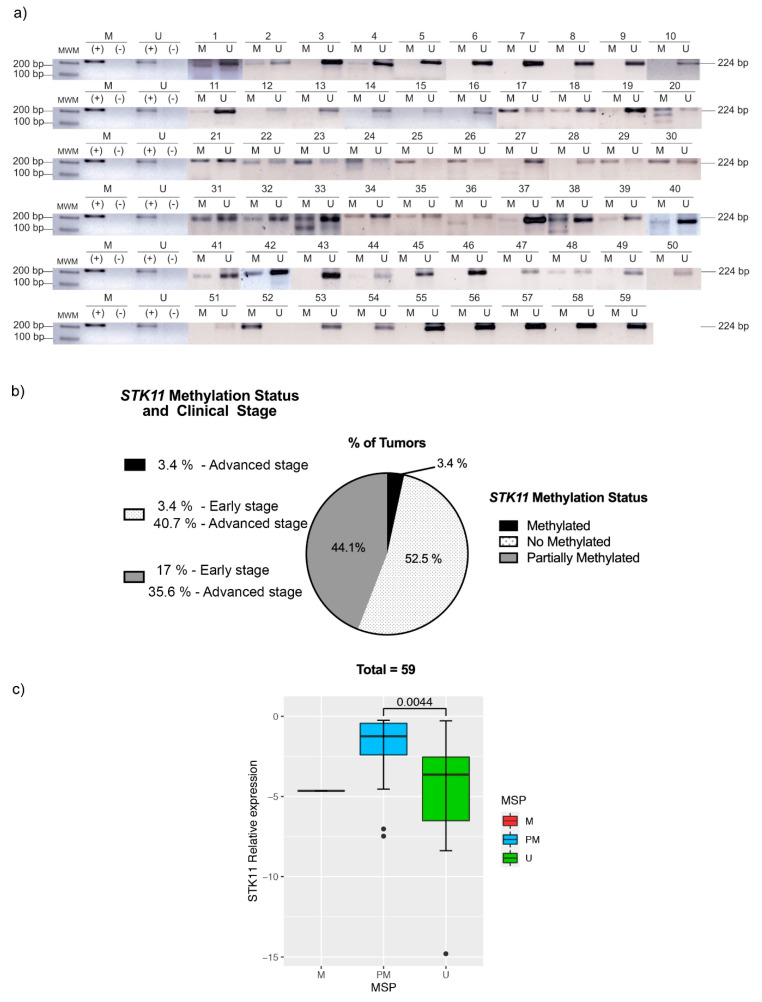
Methylation status of *STK11* assessed by MSP-PCR in patients with HNC. (**a**) End-point PCR of each sample, and (**b**) the proportion of methylated versus non-methylated and partially methylated samples. M: methylated, U: unmethylated, (−): negative control, (+): positive control, as well as the clinical stage per group. (**c**) Boxplot of the relative expression of STK11 between the MSP groups.

**Figure 3 genes-11-01058-f003:**
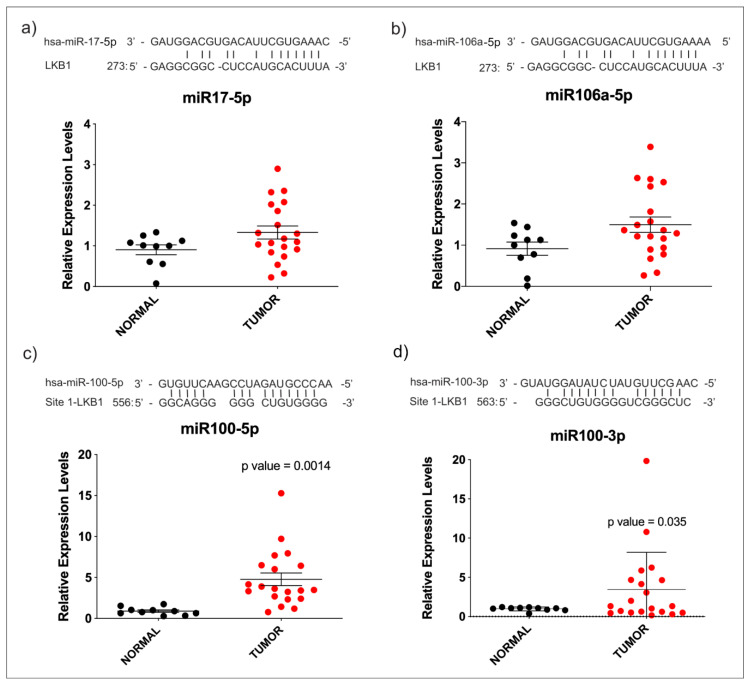
miRNAs relative expression in HNC patients. Relative expression of miR-17-5p (**a**), miR-106a-5p (**b**), miR-100-5p (**c**) and miR-100-3p (**d**) in HNC samples versus normal adjacent tissue. Seed regions of 3′-UTR of *STK11* are depicted for each miRNA (upper sequence). The statistically significant difference is indicated by *p*-values.

**Figure 4 genes-11-01058-f004:**
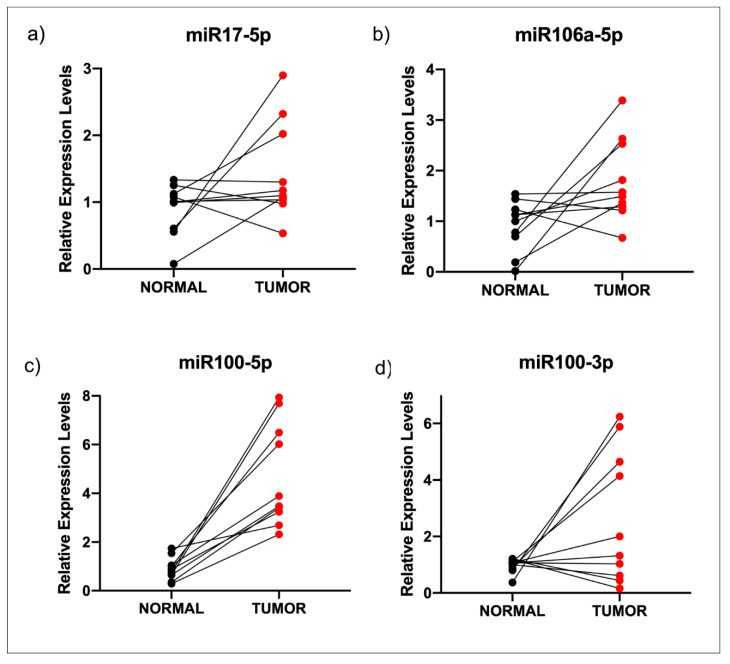
Relative expression of miRNAs in ten matched normal and carcinoma samples. Relative expression of miR-17-5p (**a**), miR-106a-5p (**b**), miR-100-5p (**c**) and miR-100-3p (**d**). As observed, only miR100-3p and -5p were overexpressed in tumor samples.

**Figure 5 genes-11-01058-f005:**
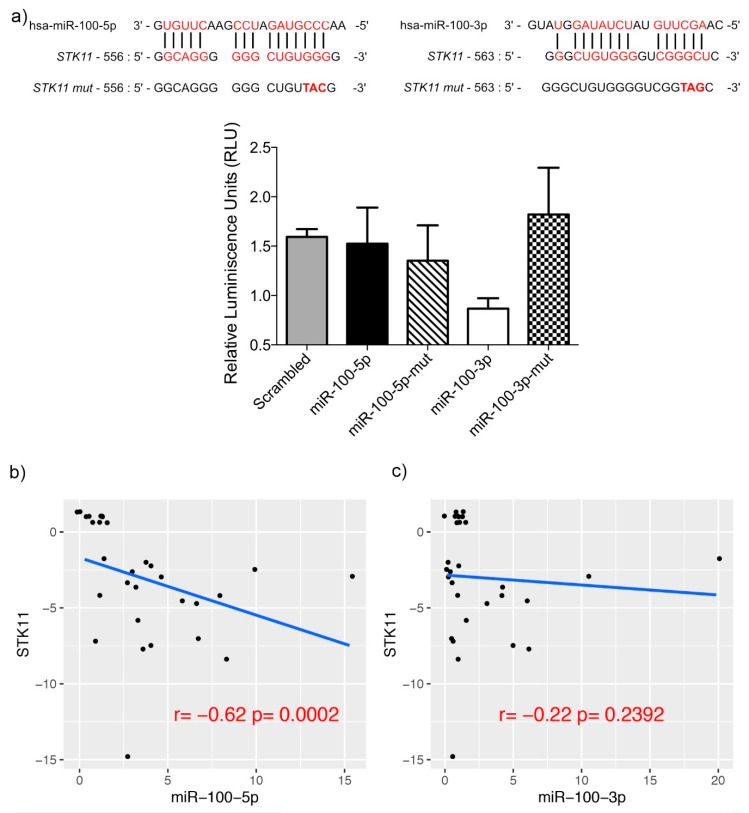
Dual-light Luciferase Assay. (**a**) Luciferase assay and constructions with the interaction of *STK11* and its mutated region with miR-100-5p and miR-100-3p. (**b**,**c**) Correlation between the expression of matched tumor samples of STK11 and miR-100-5p and miR-100-3p, respectively.

**Figure 6 genes-11-01058-f006:**
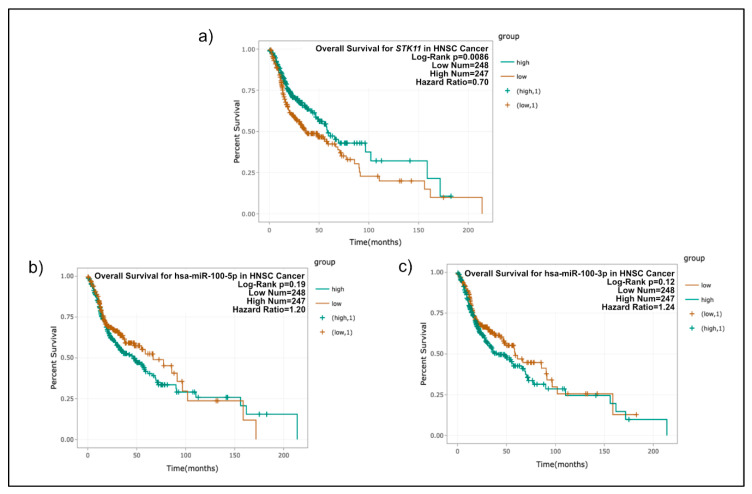
Overall survival TCGA analysis of (**a**)*STK11,* (**b**) miR-100-5p, and (**c**) miR-100-3p in head and neck cancer. The expression values of genes from RNA-seq data were scaled with log2 (FPKM + 0.01).

**Table 1 genes-11-01058-t001:** Sequences of type-specific nested PCR primers.

Primer/Cocktail	HPV Genotype	Amplicon (bp)	Sequence (5′-3′)
GP-E7-5B			CTG AGC TGT CAR NTA ATT GCT CA
GP-E6-3F		630	GGG WGK KAC TGA AAT CGG T
GP-E7-6B			TCC TCT GAG TYG YCP AAT TGC TC
Cocktail I	16	457	Forward CAC AGT TATGCA CAG AGC TGC Reverse CAT ATA TTC ATG CAA TGT AGG TGT A
	18	322	Forward CAC TTC ACT GCA AGA CAT AGA Reverse GTT GTG AAA TCGTCGTTT TTC A
	31	263	Forward GAA ATT GCA TGA ACT AAG CTC G Reverse CAC ATA TAC CTT TGTTTG TCA A
	59	215	Forward CAA AGG GGA ACT GCA AGA AAG Reverse TAT AAC AGC GTA TCA GCA GC
	45	151	Forward GTG GAA AAG TGC ATT ACA GG Reverse ACC TCT GTG GGT CCC AAT GT
Cocktail II	33	398	Forward ACT ATA CAC AAC ATT GAA CTA Reverse GTT TTT ACA CGT CAC AGT GCA
	6/11	334	Forward TGC AAG AAT GCA CTG ACC AC Reverse TGC ATG TTG TCC AGC AGT GT
	58	274	Forward GTA AAG TGT GCT TAC GAT TGC Reverse GTTGTTACA GGT TAC ACT TGT
	52	229	Forward TAA GGC TCG AGT GTG TGC AG Reverse CTAATA GTT ATT TCA CTT AAT GGT

**Table 2 genes-11-01058-t002:** Clinicopathologic characteristics of patients.

Clinical Parameters	Patients *n* = 59 (100%)	*STK11* Promoter Methylation *n* = 59	*STK11* Expression *n* = 36	miRNA Expression *n* = 20
**Gender**				
Male	40 (67.8%)			
Female	19 (32.2%)			
**Age**				
<50	10 (16.9%)			
≥50	49 (83.1%)			
**Clinical stage**				
II	5 (8.5%)	5 (8.5%)	4 (11.1%)	2 (10%)
III	11 (18.6%)	11, 3a (18.6%)	6, 3a (16.7%)	4, 3a (20%)
IVA	28 (47.5%)	28, 6a (47.5%)	15, 6a (41.7%)	8, 6a (40%)
IVB	10 (16.9%)	10 (16.9%)	8 (22.2%)	4 (20%)
IVC	5 (8.5%)	5, 1a (8.5%)	3, 1a (8.3%)	2, 1a (10%)
**Anatomic region**				
Lip and oral cavity	50 (84.7%)			
Larynx	1 (1.7%)			
Pharynx	7 (11.9%)			
Nasal cavity	1 (1.7%)			
**Histologic grade**				
Low	10 (16.9%)			
Moderate	48 (81.4%)			
High	1 (1.7%)			
**High-risk HPV**				
16	4 (6.8%)			
Negative for HPV	55 (93.2%)			
**Smoking Habit**				
Positive	34 (57.6%)			
Negative	25 (42.4%)			
**Alcoholism**				
Positive	34 (57.6%)			
Negative	25 (42.4%)			
**Tumor size**				
>5 cm	29 (49.2%)			
<5 cm	30 (50.8%)			

a Number of matched healthy adjacent tissue used as control.

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
