# Peer review of "Negative Regulation of Serine Threonine Kinase 11 (STK11) through miR-100 in Head and Neck Cancer"

_genes, 2020, doi:10.3390/genes11091058_

Round 1
Reviewer 1 Report
In this study, the authors describe a potential role for microRNA 100 in head & neck cancer (HNC) through its regulation of the serine threonine kinase 11 (STK11) that is considered a tumor suppressor. The authors demonstrate that STK11 mRNA levels are reduced in HNC tumor tissues compared to the adjacent normal tissue. Methylation-specific PCR experiments showed that downregulation of STK11 is unlikely to be due to promoter methylation. Analysis of HNC tumors samples showed increased levels of miR-100-3p and miR-100-5p. Using dual-luciferase assays, the authors demonstrate that miR-100-3p targets the 3’ region of STK11 mRNA and reduces its levels. Analysis of the HNC specimens in the TCGA database by the authors showed that higher levels of STK11 were associated with better outcome of the patients, while miR-100-3p and miR-100-5p levels were not associated with patients’ outcome. Based on their studies, the authors suggest that miR-100 may act as an oncogenic miRNA in HNC by downregulating the levels of STK11. This study contributes to our understanding of the molecular pathways that may be involved in the development of HNC.
Other Comments:
1. Why were 60 HNC tumor samples used in the study but only 10 controls (adjacent healthy tissue)?
2. How many HNC tumor samples were used to measure STK11 and miR-100 levels (Fig. 1 and Fig. 3)? It appears to be less than 60.
3. About 85% of the HNC tumor samples were from the oral cavity; Were the TCGA samples analyzed from the oral cavity or included other sites as well? TCGA data from the oral cavity may be more relevant for the current study.
4. A previous study (reference 53) showed that miR-100 is downregulated in HNC tumors and cell lines, and does not reduce the expression of STK11 mRNA in HNC cell lines transfected with this miRNA. There is no discussion of these contradictory findings in the manuscript.
Author Response
To Editor and Reviewers,
In the new version of the submitted manuscript we have taken all the comments and criticisms kindly made by the reviewers. In this new version it was of great importance the participation of Dr. Miguel Rodríguez-Morales, who has been added as author. Dr. Rodríguez-Morales has participated in the current study from the beginning; however, he was not included in the list of authors due to an error made by the corresponding author.
Thanks in advance for your kind response.
Reviewer 1
In this study, the authors describe a potential role for microRNA 100 in head & neck cancer (HNC) through its regulation of the serine threonine kinase 11 (STK11) that is considered a tumor suppressor. The authors demonstrate that STK11 mRNA levels are reduced in HNC tumor tissues compared to the adjacent normal tissue. Methylation-specific PCR experiments showed that downregulation of STK11 is unlikely to be due to promoter methylation. Analysis of HNC tumors samples showed increased levels of miR-100-3p and miR-100-5p. Using dual-luciferase assays, the authors demonstrate that miR-100-3p targets the 3’ region of STK11 mRNA and reduces its levels. Analysis of the HNC specimens in the TCGA database by the authors showed that higher levels of STK11 were associated with better outcome of the patients, while miR-100-3p and miR-100-5p levels were not associated with patients’ outcome. Based on their studies, the authors suggest that miR-100 may act as an oncogenic miRNA in HNC by downregulating the levels of STK11. This study contributes to our understanding of the molecular pathways that may be involved in the development of HNC.
Other Comments:
- Why were 60 HNC tumor samples used in the study but only 10 controls (adjacent healthy tissue)?
Response: Thanks for your kind comment. The availability of non-tumoral or apparently healthy samples is challenging from a pathological perspective. The 10 healthy samples of adjacent tissue were included based on the absence of histopathological alterations.
- How many HNC tumor samples were used to measure STK11 and miR-100 levels (Fig. 1 and Fig. 3)? It appears to be less than 60.
Response: Due to an error in the writing of the manuscript we did not specify that 36 samples were used for the expression of STK11; while 20 tumor samples were used for the analysis of the three microRNAs. Respecting to 10 non-tumoral adjacent tissue were employed as controls in both kind of experiments. It should be explained that this was mainly due to the limited abundance of total RNA in the samples used in this study. This omission was included in material and methods P5 L153-154 and 167-168.
- About 85% of the HNC tumor samples were from the oral cavity; Were the TCGA samples analyzed from the oral cavity or included other sites as well? TCGA data from the oral cavity may be more relevant for the current study.
RESPONSE: Thank you for your comment. We used all the patients from the TCGA Head and Neck project which was composed of 295 patients with lip and coral cavity (59.59%), 111 of Larinx (22.4%) and 89 of Pharynx (17.97%). We opted to analyze all the tumors to cover a wider patient cohort when assessing the clinical relevance of the molecules.
- A previous study (reference 53) showed that miR-100 is downregulated in HNC tumors and cell lines, and does not reduce the expression of STK11 mRNA in HNC cell lines transfected with this miRNA. There is no discussion of these contradictory findings in the manuscript.
RESPONSE: We made a discussion about discrepancies found in both studies P13 L353-359
Reviewer 2 Report
The manuscript by Figueroa-González et al describes an comprehensive evaluation of the effects of mir-100 on the expression of the STK11 gene LKB1 protein.
The study is robust, methodologically comprehensive and complemented with TCGA data reanalysis. However, the description of the methods, presentation of the results and parts of the discussion need to be significantly improved to add clarity and avoid misinterpretation. Some of the relevant results or information is not show.
The specific points needing clarification are listed below in a Page – Line format (ie P3 L97 stands for page 3 line 97)
Inconsistencies or unclarities
P3 L97 states that eligibility (inclusion?) criteria were “confirmed pathological diagnosis of head and neck cancer stages II to IV”. However, Table 2 has one patient with Tis also Table 2 lists “Clinical stage” instead of pathohistological diagnosis. Which staging scheme was used? AJCC 8th edition includes the HPV data in staging. Was this also considered or not?
P6 L196 additional variables listed in the statistical analysis section were smoking habit, alcoholism, tumor size, and clinical stage. Table 2 contains “Histologic grade” as well. Was this variable intentionally omitted? The variables of smoking and alcohol use are not defined? Is it really clinically diagnosed alcoholism listed in the table or is this just some level of daily/weekly alcohol use and if so what was considered as the cutoff. What was considered as “Tabaquism” any smoking history or again some cutoff?. Table 2 reveals that tumors within 2 anatomic locations (larynx and nasal cavity) were only observed once each. How do the n=1 groups influence multivariate analysis if it was performed as implied by “including covariates in the model” (P6L196)?
P7 L212 which patients do the 10 non-pathological controls come from? How representative are they of the population shown in Table 2.The same question could be asked about the samples selected for subsequent analysis. How those samples compare to the whole set of 60 patients? Possibly Table 2 could be expanded to include the same information for 42 samples tested for mRNA/methylation/miRNA and another column to provide the data on 10 normal tissue” controls. Table 2 could also be expanded with HPV information across the sample groups.
P7 L214 The text describing the expression level analysis is inconsistent with the table 2 and possibly figure 1. Foremost, the text is difficult to read but critically implies that 6 samples of stage II were tested while table 2 lists only 5 stage II tumors. Also the text implies 7 + 28 +1 +6 tumors were tested while only 19 red squares are shown. When selecting samples for subsequent analyses only the tumor stage was considered. However, it is well known that oral and oropharyngeal sites are quite different with respect to HPV involvement. Were the samples selected for subsequent analysis of oral origin or mixed oral/oropharyngeal origin and are there differences in the two regions? To avoid uncertainty it would be best if it was made clear throughout the manuscript that the data provided mostly refer to oral cancer if only a small number of oropharyngeal cases were included.
P7 L222 Figure 1 legend is not in line with the figure. No grey bar or black bar can be seen. The graph appears to be showing all sample points as well as mean + SD horizontal line while the legend implies a bar graph? The stars present on the graph *** are not explained nor needed since actual p value is shown. The legend duplicates the p<0.0001 information unnecessarily. Y axis should include the value of “1” since this is the level the fold change is shown against? Was an average of all control measurements calculated and all samples shown in relation to this average? If so, only one point should be shown for normal samples. If each control was paired with its respective tumor sample, than only 10 such samples could be shown on this graph. Figure shows log transformed values but neither the methods nor the y-axis label nor the figure legend explains it sufficiently. Figure 1 could easily be expanded to provide additional data on the low/high stage tumors as well as TCGA data as external validation
P8 L235 Figure 2. What was used for positive and negative controls. Percentages of groups according to the methylation status would be useful to show. Was methylation in normal tissues assessed? Figure 2b could easily be expanded to cover the low and high stage tumor data
P8 L243 which databases? Tarbase which includes data on experimentally validated miRNA–gene interactions lists more candidates than those shown in the manuscript (http://carolina.imis.athena-innovation.gr/diana_tools/web/index.php?r=tarbasev8%2Findex&miRNAs[]=&genes[]=STK11&query=)
P8 L244. For the sake of completeness, for all miRNA tested the -3p or -5p end should be indicated. For miR-106 it should also be indicated whether the 106a or 106b was tested throughout the manuscript and not only at some places.
P8 L247 it is not completely clear which samples were tested for miRNA expression. Only matched pairs or again more selected samples across different stages of the HNC
P9 L250 Figure 3 also lacks the mention of log transformation
P9 L263-265 the last sentence of this paragraph seems out of place. miRNA detection was done in patient samples collected by the authors, while Kaplan meier analysis was done on TCGA data. Also, the abstract and methods mention that the expression levels from TCGA data were analyzed. However this data is not shown and should be. Do the TCGA data confirm the finding of lower STK11 expression and higher miRNA expression as in the 60 clinical patient samples? This information could easily be presented and would strengthen the manuscript greatly while the Kaplan meier plots are not so much informative.
P11 L280-287. For the types common in HNSCC please refer to the publication Kreimer AR, Clifford GM, Boyle P, Franceschi S. Human papillomavirus types in head and neck squamous cell carcinomas worldwide: a systematic review. Cancer Epidemiol Biomarkers Prev. 2005;14(2):467-475. doi:10.1158/1055-9965.EPI-04-0551. Finding only HPV16 in the study is not unusual since only 4 samples were positive. The sentence „particular distribution of high- and low-risk HPVs in the Mexican population,“ is misleading since only HPV6/11 was tested and not found. Since only 4 samples were positive, it is not opportune to speculate about particular distributions as 4 samples are not enough to establish the pattern or claim this “pattern” differs from current knowledge (which it doesn’t)
P11 L305-307 the sentences are inconsistent. In one sentence the authors strongly imply that promotor methylation is not influencing STK11 downregulation, while in the other the authors imply that the methylation could be influencing STK11 downregulation. Possibly the methodology could be discussed (https://www.ncbi.nlm.nih.gov/pmc/articles/PMC3414395/)
P11 L 317 there is no data shown regarding the methylation and HPV presence. It is also not clearly stated whether HPV positive tumours were selected for the STK11 mRNA or promotor methylation analysis.
P11 L321 No data is presented regarding the MSP testing and clinical stage, yet such data could be easily shown since Figure 2b has significant amount of empty or uninformative space
P12 L330-335 The data suggests that mir-100-5p doesn’t affect the luciferase assay to a great extent. Mir-100-3p does appear to have a strong effect at least in the luciferase assay. However, according to miRBASE database, mir-100-5p is the dominant form of this miR and the -3p is barely if at all expressed (5 million reads versus 270 reads in NGS experiments for -5p and -3p, respectively; http://www.mirbase.org/cgi-bin/mirna_entry.pl?acc=MI0000102). Since the terminology is inconsistently used it cannot clearly be know which version authors use when saying only mir-100, however previously mir-100 referred to what is now known as mir-100-5p while mir-100* referred to miR-100-3p. When discussing previous studies by other authors care should be taken to clearly show which miR is compared to which so that miR-100-3p results are not inadvertently and erroneously put in context of miR-100-5p previous data.
Also extra care should be taken when discussing miRNAs so that subsequent authors clearly know which molecule the data is presented for.
Trivial problems and typos
P3 L 108 „2.3 HVP…“ should be HPV
P3 L120-121 Instead of primer sequences for beta globin, a reference would be better as it would cover the conditions as well as the primers. If the primers are novel, then the conditions need to be listed as well.
P3 L123-125 the text implies no negative controls were present?
P5 L154, Additional information on RNAhybrid 2.2 is needed, a website, reference and/or a brief summary of the method is needed.
P5 L158 typo in transcription „TaqMan Micro-RNA Reverse Transcrtiption Kit“
P6 L186-187 „OS“ abbreviation was not defined before
P6 L195 including covariates implies that multivariate models were made. The text implicitly states that only univariate analysis was performed however.
P11 L278 Typo “VPH-infected”
P11 L293 typos? “activating AMPK; 12 then, LKB1/AMPK”
Author Response
To Editor and Reviewers,
In the new version of the submitted manuscript we have taken all the comments and criticisms kindly made by the reviewers. In this new version it was of great importance the participation of Dr. Miguel Rodríguez-Morales, who has been added as author. Dr. Rodríguez-Morales has participated in the current study from the beginning; however, he was not included in the list of authors due to an error made by the corresponding author.
Thanks in advance for your kind response
Reviewer 2
The specific points needing clarification are listed below in a Page – Line format (ie P3 L97 stands for page 3 line 97)
Inconsistencies or unclarities
P3 L97 states that eligibility (inclusion?) criteria were “confirmed pathological diagnosis of head and neck cancer stages II to IV”. However, Table 2 has one patient with Tis also Table 2 lists “Clinical stage” instead of pathohistological diagnosis. Which staging scheme was used? AJCC 8th edition includes the HPV data in staging. Was this also considered or not?
RESPONSE: Thanks for your kind comment. There was an error including the patients and one of them was incorrectly classified. It has been removed from the table and the text; therefore 59 patients were finally included. The AJCC 8th edition was used to classify included patients. HPV determination was performed by PCR and considered to this aim.
P6 L196 additional variables listed in the statistical analysis section were smoking habit, alcoholism, tumor size, and clinical stage. Table 2 contains “Histologic grade” as well. Was this variable intentionally omitted? The variables of smoking and alcohol use are not defined? Is it really clinically diagnosed alcoholism listed in the table or is this just some level of daily/weekly alcohol use and if so what was considered as the cutoff. What was considered as “Tabaquism” any smoking history or again some cutoff?. Table 2 reveals that tumors within 2 anatomic locations (larynx and nasal cavity) were only observed once each. How do the n=1 groups influence multivariate analysis if it was performed as implied by “including covariates in the model” (P6L196)?
RESPONSE: Table 2 has been corrected as requested. Smokers were defined by those subjects who consumed more than 15 cigarettes a day for a period of time ≥15 years. Alcohol drinking as consumers of ≥4 whisky equivalents (30 cc for each equivalent) daily for ≥15 years, included in the text, P7 L223-225. Respecting to multivariate analysis, we omitted to clarify that those variables less than three cases were excluded. In the revised version P6 L214 is included.
P7 L212 which patients do the 10 non-pathological controls come from? How representative are they of the population shown in Table 2. The same question could be asked about the samples selected for subsequent analysis. How those samples compare to the whole set of 60 patients? Possibly Table 2 could be expanded to include the same information for 42 samples tested for mRNA/methylation/miRNA and another column to provide the data on 10 normal tissue” controls. Table 2 could also be expanded with HPV information across the sample groups.
RESPONSE: We have added the information of samples of each experiment and the information of adjacent healthy tissue for them; also, it was included the HPV data as requested. Also we have added in figure 1 a panel b in which STK11 expression level is analyzed in tumor and normal matched adjacent tissue. The same was done for microRNA expression data, hence a figure 4 was added in the original manuscript.
P7 L214 The text describing the expression level analysis is inconsistent with the table 2 and possibly figure 1. Foremost, the text is difficult to read but critically implies that 6 samples of stage II were tested while table 2 lists only 5 stage II tumors. Also the text implies 7 + 28 +1 +6 tumors were tested while only 19 red squares are shown. When selecting samples for subsequent analyses only the tumor stage was considered. However, it is well known that oral and oropharyngeal sites are quite different with respect to HPV involvement. Were the samples selected for subsequent analysis of oral origin or mixed oral/oropharyngeal origin and are there differences in the two regions? To avoid uncertainty it would be best if it was made clear throughout the manuscript that the data provided mostly refer to oral cancer if only a small number of oropharyngeal cases were included.
RESPONSE: The clinical and molecular data were carefully analyzed to avoid any incorrect or erroneous interpretation, so in the P8 and P9 L236 – 239, TABLE 2 AND FIG 1 we include the kind observations of the reviewer.
P7 L222 Figure 1 legend is not in line with the figure. No grey bar or black bar can be seen. The graph appears to be showing all sample points as well as mean + SD horizontal line while the legend implies a bar graph? The stars present on the graph *** are not explained nor needed since actual p value is shown. The legend duplicates the p<0.0001 information unnecessarily. Y axis should include the value of “1” since this is the level the fold change is shown against? Was an average of all control measurements calculated and all samples shown in relation to this average? If so, only one point should be shown for normal samples. If each control was paired with its respective tumor sample, than only 10 such samples could be shown on this graph. Figure shows log transformed values but neither the methods nor the y-axis label nor the figure legend explains it sufficiently. Figure 1 could easily be expanded to provide additional data on the low/high stage tumors as well as TCGA data as external validation
RESPONSE: Thank you for your comments, Figure 1, has been modified by adding all your observations. The expression level of each normal adjacent sample The relative expression data were included to correlate the adjacent healthy tissue samples with their respective tumor. A new figure 1 panel b shows the expression for STK11 in matched normal-tumor samples. TCGA data were used to analyze the clinical relevance of the expression values in the overall survival.
P8 L235 Figure 2. What was used for positive and negative controls. Percentages of groups according to the methylation status would be useful to show. Was methylation in normal tissues assessed? Figure 2b could easily be expanded to cover the low and high stage tumor data
RESPONSE: Thank you for your observation. Human Methylated and Non-methylated DNA sets were used as a positive and negative control, respectively, and this information was included in material and methods P3 L138, 139. The methylation in normal samples was not analyzed, mainly due to the high amount of DNA needed for each experiment. Figure 2 has been modified to show the percentage of methylation status and tumor stage for each group, P10.
P8 L243 which databases? Tarbase which includes data on experimentally validated miRNA–gene interactions lists more candidates than those shown in the manuscript (http://carolina.imis.athena-innovation.gr/diana_tools/web/index.php?r=tarbasev8%2Findex&miRNAs[]=&genes[]=STK11&query=)
RESPONSE: Thank you for your comment. We used the miRWalk2.0 and RNAHybrid 2.2 platforms to perform the target prediction. We added this information in the materials and methods section.
P8 L244. For the sake of completeness, for all miRNA tested the -3p or -5p end should be indicated. For miR-106 it should also be indicated whether the 106a or 106b was tested throughout the manuscript and not only at some places.
RESPONSE: Thanks for your observation, throughout the entire text, the miRNAs analyzed were specified.
P8 L247 it is not completely clear which samples were tested for miRNA expression. Only matched pairs or again more selected samples across different stages of the HNC.
RESPONSE: Thank you for your comments. The relative expression data were included to correlate the adjacent healthy tissue samples with their respective tumor for miRNA expression; Table 2 shows the number of healthy tissue samples for each clinical stage used for miRNA and Luciferase reporter assay. A new figure has been included in which the expression values for each miRNA in matched normal-tumor samples are shown.
P9 L250 Figure 3 also lacks the mention of log transformation
There is no log Transformation. All values are relative units versus normal samples.
P9 L263-265 the last sentence of this paragraph seems out of place. miRNA detection was done in patient samples collected by the authors, while Kaplan meier analysis was done on TCGA data. Also, the abstract and methods mention that the expression levels from TCGA data were analyzed. However this data is not shown and should be. Do the TCGA data confirm the finding of lower STK11 expression and higher miRNA expression as in the 60 clinical patient samples? This information could easily be presented and would strengthen the manuscript greatly while the Kaplan meier plots are not so much informative.
RESPONSE: Thank you for your comment, we added this analysis in lines 290-295 of the manuscript. We observed that the expression of STK11 was highly heterogenous in tumor samples and although we observed no significant differences in the expression of STK11 between normal vs tumor tissues, the Kaplan Meier analysis indicated that tumors showed different survival curves when they are classified in high and low groups. To our understanding, the ability of this molecule to be clinically relevant in tumors independent of its expression in normal tissues is of highly importance.
P11 L280-287. For the types common in HNSCC please refer to the publication Kreimer AR, Clifford GM, Boyle P, Franceschi S. Human papillomavirus types in head and neck squamous cell carcinomas worldwide: a systematic review. Cancer Epidemiol Biomarkers Prev. 2005;14(2):467-475. doi:10.1158/1055-9965.EPI-04-0551. Finding only HPV16 in the study is not unusual since only 4 samples were positive. The sentence „particular distribution of high- and low-risk HPVs in the Mexican population,“ is misleading since only HPV6/11 was tested and not found. Since only 4 samples were positive, it is not opportune to speculate about particular distributions as 4 samples are not enough to establish the pattern or claim this “pattern” differs from current knowledge (which it doesn’t)
RESPONSE: Thank you for your comments. The publication of Kreimer et al. is already included in the manuscript. Despite in our work we just found four HPV-16 positive samples, we were also searching for more genotypes (Two primer cocktails were used for genotyping: cocktail 1 for HPV-16, -18, -31, -59, and -45 and cocktail 2 for HPV-33, -6/11, -58, -52, and -56).
P11 L305-307 the sentences are inconsistent. In one sentence the authors strongly imply that promotor methylation is not influencing STK11 downregulation, while in the other the authors imply that the methylation could be influencing STK11 downregulation. Possibly the methodology could be discussed (https://www.ncbi.nlm.nih.gov/pmc/articles/PMC3414395/)
RESPONSE: Thank you for your observations. A mistake was made in the text, and it was corrected. In our study, there was no clear evidence that promoter methylation status is related to STK11 downregulation. The reference you kindly gave us, was included in the discussion. P13 L-338-340.
P11 L 317 there is no data shown regarding the methylation and HPV presence. It is also not clearly stated whether HPV positive tumors were selected for the STK11 mRNA or promotor methylation analysis.
RESPONSE: Thanks for the observation. The four positive HPV-16 samples were used for STK11 expression, miRNAs expression and MSP assay. This information was included in Results P7, L-230
P11 L321 No data is presented regarding the MSP testing and clinical stage, yet such data could be easily shown since Figure 2b has significant amount of empty or uninformative space
RESPONSE: Thanks for your comment. Figure 2b, was modified in order to show the percentage of methylation status and clinical stage for each group, P10
P12 L330-335 The data suggests that mir-100-5p doesn’t affect the luciferase assay to a great extent. Mir-100-3p does appear to have a strong effect at least in the luciferase assay. However, according to miRBASE database, mir-100-5p is the dominant form of this miR and the -3p is barely if at all expressed (5 million reads versus 270 reads in NGS experiments for -5p and -3p, respectively; http://www.mirbase.org/cgi-bin/mirna_entry.pl?acc=MI0000102). Since the terminology is inconsistently used it cannot clearly be know which version authors use when saying only mir-100, however previously mir-100 referred to what is now known as mir-100-5p while mir-100* referred to miR-100-3p. When discussing previous studies by other authors care should be taken to clearly show which miR is compared to which so that miR-100-3p results are not inadvertently and erroneously put in context of miR-100-5p previous data. Also extra care should be taken when discussing miRNAs so that subsequent authors clearly know which molecule the data is presented for.
RESPONSE: Since miR-100-5p is the dominant form, in the discussion, we assumed that other authors refer to miR-100-5p when they write miR-100. In the manuscript, it was corrected all miR-100-3p and -5p instead of just miR-100.
Trivial problems and typos
P3 L 108 „2.3 HVP…“ should be HPV
RESPONSE: It was corrected.
P3 L120-121 Instead of primer sequences for beta globin, a reference would be better as it would cover the conditions as well as the primers. If the primers are novel, then the conditions need to be listed as well.
RESPONSE: The conditions of amplification for beta globin, were included in the text. P3, L120-124.
P3 L123-125 the text implies no negative controls were present?
RESPONSE: Thank you for your observation, it was omitted in the text, but it was used a negative control as well as a positive control for each PCR. It as corrected. P3. L125
P5 L154, Additional information on RNAhybrid 2.2 is needed, a website, reference and/or a brief summary of the method is needed.
RESPONSE: Thank you, this observation was included in the text. P5, L158-160
P5 L158 typo in transcription „TaqMan Micro-RNA Reverse Transcrtiption Kit“
RESPONSE: It was corrected.
P6 L186-187 „OS“ abbreviation was not defined before
RESPONSE: Thank you for your comment, it was corrected in P6, L203.
P6 L195 including covariates implies that multivariate models were made. The text implicitly states that only univariate analysis was performed however.
RESPONSE: There was a mistake in the Statistical analysis description; the word univariate was changed by multivariate.
P11 L278 Typo “VPH-infected”
RESPONSE: It was corrected.
P11 L293 typos? “activating AMPK; 12 then, LKB1/AMPK”
RESPONSE: It was corrected.
Round 2
Reviewer 1 Report
The authors have responded appropriately to my comments and the manuscript is improved.
Author Response
There are no comments about our previous report
Reviewer 2 Report
The revised manuscript by Figueroa-González et al. improves on the previous version and clarifies many of the issues found therein. However, some issues still remain unclear or ignored
To summarize the study, the authors confirmed that downregulation of STK11 expression is important in tumors (and in patient survival from TCGA data). However, the downregulation was not associated with promoter methylation. The authors then assessed miRNA regulation and found that miR100-5p (dominant form) is increased robustly but doesn’t influence STK11 expression in luciferase test. The major luciferase effect was seen with 100-3p (yet this from is weakly expressed overall) and only slightly overexpressed in their samples (p=0.035). Only 4 of 10 matched samples exhibit overexpression (Fig 4d) and 8/20 overall (fig 3d)
Taken in this context the last 2 paragraphs of the discussion as well as the conclusions on page 15 are factually correct but incomplete. Since mir100-3p is not the dominant form, it remains unlikely that mir100-3p is solely responsible for the expression reduction observed.
The authors could add more credibility and impact by using and presenting more of their results. The described statistical analysis part of the methods section still has no clear link to any results shown in the results or discussion sections? In most cases the data which would be assessed by statistical tests remains "not shown" with no p values provided even those above p=0.05. The multivariate model remains completely not shown or mentioned beyond the methods section. It is still unclear where ANOVA was used? Figures only present comparisons of 2 groups. The ANOVA would be useful for comparing results across different stages, yet the authors failed to revise the figures and add stage information to Figs1, 2 3 in a way to justify the use of ANOVA test?
Also instead of making a model for methylation of STK11 promotor, it would be immensely more interesting to see the associations of variables with the STK11 mRNA expression even if only on 20 samples. Also, instead of omitting variables, it is more appropriate to group samples. For example, instead of omitting High histologic grade n=1 it would be better to make a Moderate+High histologic group with N=49. With anatomic location maybe 2 groups can be made “Lip and oral cavity” n=50 and “Other” N=9 instead of completely removing 2 cases each with 1 sample (larynx and nasal cavity). Possibly there Is a better and clinically more meaningful way to make groups?
At least testing the correlation associations of STK11 expression to age or stage or even miRNA expression might strengthen the results?
Furthermore, the authors in the methods section introduce a new heading “TCGA data analysis”, however the results of the described Deseq2 analysis are also not presented in the manuscript? This heading does not follow the numbering of other paragraphs in the methods section?
The table 2 still refers to tabaquism while in the rest of the manuscript it is "smoking habit"
For figure 2, the results for stages could be presented grouped as in the text on “early or advanced stages“. Thus instead of stating “No association was found between HNC clinical stages and STK11 demethylated or partially methylated status, as previously mentioned (data are not shown). “ the authors could show this data and the result and name of the statistical test used for this claim.
The same can be said for Figure 1 where instead of stating “Furthermore, no significant difference was found in the STK11 expression level in early or advanced stages of HNC tumors (data are not shown),“ the actual data could easily be shown along with the statistical testing result with ANOVA. For example, relative expression in normal, early stage and advanced stage groups could be presented.
Table 2, instead of inappropriately adding rows at the bottom of the table, the samples selected for STK11 expression (n=36) and miRNA expression (n=20) and healthy control tissue (n=10) should have been added as 3 additional columns right of the "Patients n=59 (100%)" column. The table 2 contains the “Histology” variable which is in fact not variable with all cases being SCC. This data is better shown in text since it is a feature of study design and doesn’t need presenting across groups.
Also the revised table 2 shows that the few HPV positive samples are of the increasing importance in different subsets of samples. Of all 59 samples 4 were HPV positive (6.7%), however, 4/36 (11.1%) and 4/20 (20%) samples used for mRNA and miRNA analysis are HPV positive which might overinflate the HPV contribution in an otherwise oral cancer context where HPV involvement is limited.
Table 2 “high risk HPV” variable inappropriately contains the low risk types 6 and 11. Also instead of listing all types tested, it might be more appropriate to replace his with “Other HPV types” and possibly list them as a footnote.
Figure 4d is interesting since there were 4 tumors where there was a large increase of mir100-3p and 6 tumors where there was almost no increase or even a decrease of mir100-3p. Is there some feature that can distinguish the two groups? Maybe the 4 HPV positive samples are the ones with marked 100-3p increase? Similarly on Fig 3d there is a clear separation of tumors with mir100-3p expression almost identical to normal and 8/20 where there is a marked increase. Is it possible to differentiate those 2 groups based on some clinical parameter?
On page 12 the TCGA results still share the paragraph with luciferase assay. Since the content is completely different, the TCGA results should be separated in a new paragraph.
Some English language polishing is still required, as the new text introduced some new language errors. For example the following (but other text needs to be thoroughly checked as well):
Page 3 "in the same thermal cycler than the first PCR"
Page 5 "normal adjacent samples were employed"
Page 6 "based in the median of"
Author Response
The revised manuscript by Figueroa-González et al. improves on the previous version and clarifies many of the issues found therein. However, some issues still remain unclear or ignored
To summarize the study, the authors confirmed that downregulation of STK11 expression is important in tumors (and in patient survival from TCGA data). However, the downregulation was not associated with promoter methylation. The authors then assessed miRNA regulation and found that miR100-5p (dominant form) is increased robustly but doesn’t influence STK11 expression in luciferase test. The major luciferase effect was seen with 100-3p (yet this from is weakly expressed overall) and only slightly overexpressed in their samples (p=0.035). Only 4 of 10 matched samples exhibit overexpression (Fig 4d) and 8/20 overall (fig 3d)
Taken in this context the last 2 paragraphs of the discussion as well as the conclusions on page 15 are factually correct but incomplete. Since mir100-3p is not the dominant form, it remains unlikely that mir100-3p is solely responsible for the expression reduction observed.
Response: Thanks for your kind comment that we employed to enhance our work. Indeed, the reduction of expression of STK11 can not be explained solely by miR-100-3p, we have modified several section of the manuscript. P1 L38-45, P2 L85-87, P16 L403-404, L408-410.
The authors could add more credibility and impact by using and presenting more of their results. The described statistical analysis part of the methods section still has no clear link to any results shown in the results or discussion sections? In most cases the data which would be assessed by statistical tests remains "not shown" with no p values provided even those above p=0.05. The multivariate model remains completely not shown or mentioned beyond the methods section. It is still unclear where ANOVA was used? Figures only present comparisons of 2 groups. The ANOVA would be useful for comparing results across different stages, yet the authors failed to revise the figures and add stage information to Figs1, 2 3 in a way to justify the use of ANOVA test?
Response: We added the results of the ANOVA for the expression of STK11, the miRNAs and the clinical stages and histological grade of the tumors as supplementary file 1, we also described in Methods section the Statistical Analysis employed P7 L202-207 and mentioned the results in subsection 3.2, P9 L251-257.
Also instead of making a model for methylation of STK11 promotor, it would be immensely more interesting to see the associations of variables with the STK11 mRNA expression even if only on 20 samples. Also, instead of omitting variables, it is more appropriate to group samples. For example, instead of omitting High histologic grade n=1 it would be better to make a Moderate+High histologic group with N=49. With anatomic location maybe 2 groups can be made “Lip and oral cavity” n=50 and “Other” N=9 instead of completely removing 2 cases each with 1 sample (larynx and nasal cavity). Possibly there Is a better and clinically more meaningful way to make groups?
Response: We grouped the clincal stages in early and advanced and analyzed the possible differences in the expression of STK11 using the clinical stage, histological grade, MSP and anatomic region using an ANOVA which can be found in the supplementary file 1.
We observed that only MSP was significant in the analysis, whereas the other clinical attributes had a higher p-value close to 1, thus merging them in groups for a second analysis would yield the same results.
At least testing the correlation associations of STK11 expression to age or stage or even miRNA expression might strengthen the results?
Response: Thank you for your comment, we added this correlation in figure 5b and c.
Furthermore, the authors in the methods section introduce a new heading “TCGA data analysis”, however the results of the described Deseq2 analysis are also not presented in the manuscript? This heading does not follow the numbering of other paragraphs in the methods section?
Response: We fixed the subheading and merged this section with subsection 2.8.
The table 2 still refers to tabaquism while in the rest of the manuscript it is "smoking habit"
Response: Changed
For figure 2, the results for stages could be presented grouped as in the text on “early or advanced stages“. Thus instead of stating “No association was found between HNC clinical stages and STK11 demethylated or partially methylated status, as previously mentioned (data are not shown). “ the authors could show this data and the result and name of the statistical test used for this claim.
Response: We performed an ANOVA which yielded no significant differences between the STK11 expression and the clinical stages, we added this information in section 3.2
The same can be said for Figure 1 where instead of stating “Furthermore, no significant difference was found in the STK11 expression level in early or advanced stages of HNC tumors (data are not shown),“ the actual data could easily be shown along with the statistical testing result with ANOVA. For example, relative expression in normal, early stage and advanced stage groups could be presented.
Response: For the normal vs tumor expression analysis we employed an two-tailed Student’s t-test, whereas for the analysis between the expression and the clinicopathological characteristics we used a two-way anova. We modified the subsection 2.9 to reflect this P6 L202-207.
Table 2, instead of inappropriately adding rows at the bottom of the table, the samples selected for STK11 expression (n=36) and miRNA expression (n=20) and healthy control tissue (n=10) should have been added as 3 additional columns right of the "Patients n=59 (100%)" column. The table 2 contains the “Histology” variable which is in fact not variable with all cases being SCC. This data is better shown in text since it is a feature of study design and doesn’t need presenting across groups.
Also the revised table 2 shows that the few HPV positive samples are of the increasing importance in different subsets of samples. Of all 59 samples 4 were HPV positive (6.7%), however, 4/36 (11.1%) and 4/20 (20%) samples used for mRNA and miRNA analysis are HPV positive which might overinflate the HPV contribution in an otherwise oral cancer context where HPV involvement is limited.
Table 2 “high risk HPV” variable inappropriately contains the low risk types 6 and 11. Also instead of listing all types tested, it might be more appropriate to replace his with “Other HPV types” and possibly list them as a footnote.
RESPONSE: Thank you for your comments, table 2, was modified according to your observations. the histology was added in the text P7 L218,219, as well as the other HPV genotypes analyzed P7, L 220-222.
Figure 4d is interesting since there were 4 tumors where there was a large increase of mir100-3p and 6 tumors where there was almost no increase or even a decrease of mir100-3p. Is there some feature that can distinguish the two groups? Maybe the 4 HPV positive samples are the ones with marked 100-3p increase? Similarly on Fig 3d there is a clear separation of tumors with mir100-3p expression almost identical to normal and 8/20 where there is a marked increase. Is it possible to differentiate those 2 groups based on some clinical parameter?
RESPONSE: Thank you for your observation. In miR100-3p expression, just two of the all overexpressed samples correspond to an HPV-6 positive sample, That is why it is not possible to establish a correlation between increased miR-100-3p and HPV positiveness.
Some English language polishing is still required, as the new text introduced some new language errors. For example the following (but other text needs to be thoroughly checked as well):
Page 3 "in the same thermal cycler than the first PCR"
Page 5 "normal adjacent samples were employed"
Page 6 "based in the median of"
Response: The text has been rigorously checked for language errors.
